# Fuzzy Adaptive Command-Filter Control of Incommensurate Fractional-Order Nonlinear Systems

**DOI:** 10.3390/e25060893

**Published:** 2023-06-02

**Authors:** Dianjun Gong, Yong Wang

**Affiliations:** Department of Automation, University of Science and Technology of China, Hefei 230052, China; gdj1005@mail.ustc.edu.cn

**Keywords:** incommensurate fractional-order systems, fuzzy adaptive control, command-filter control

## Abstract

This paper focuses on the command-filter control of nonstrict-feedback incommensurate fractional-order systems. We utilized fuzzy systems to approximate nonlinear systems, and designed an adaptive update law to estimate the approximation errors. To overcome the dimension explosion phenomenon in the backstepping process, we designed a fractional-order filter and applied the command filter control technique. The closed-loop system was semiglobally stable, and the tracking error converged to a small neighbourhood of equilibrium points under the proposed control approach. Lastly, the validity of the developed controller is verified with simulation examples.

## 1. Introduction

Fractional-order calculus is an extension of classical integer-order calculus that has had a history of several hundreds of years. Due to the lack of a practical application background, the development of fractional-order theory has stagnated for a long time. In recent decades, with the rapid development of science and technology, an increasing number of practical physical systems were found to have fractional-order characteristic [1,2,3,4]. An ocean of valuable results on linear fractional-order systems (FOSs) have been published in [5,6,7,8]. To discuss the stability of nonlinear FOSs, the direct and indirect Lyapunov methods were proposed in [9,10], respectively. Adaptive control, sliding mode control, fuzzy control, and the comprehensive application of these schemes was extended to FOSs [11,12,13,14,15]. In [16,17], Ding et al. constructed an adaptive backstepping controller for nonlinear FOSs with additive disturbance.

Most of the existing results are on the control issue of commensurate fractional-order systems. In practical applications, depicting the physical phenomenon with incommensurate fractional models is more reasonable. It was experimentally verified that the incommensurate model exhibited superior performance over its commensurate counterpart in the field of thermal diffusion dynamics and lithium-ion batteries [18,19]. Thanks to the proposal of the indirect Lyapunov method, Wei et al. developed an adaptive backstepping control approach in [20], and an output-feedback control method in [21] for incommensurate nonlinear FOSs. Sheng et al. [22] proposed an observer-based adaptive backstepping control method for nonlinear FOSs, and a fractional-order auxiliary system was constructed to compensate for the input saturation by generating a series of virtual signals. However, in the process of backstepping controller design, reference signals need to be calculated repeatedly, which leads to the problem of computational complexity explosion.

Dynamic surface control (DSC), which was first proposed in [23], is an effective approach to reduce the complexity of computation. Zhou et al. [24] proposed a DSC scheme for nonlinear systems with uncertainty, and external disturbances could be approximated with radial-basis-function neural networks whose weight value is adjusted online. Ning et al. [25] focused on the collective behaviors of robots beyond nearest-neighbor rules, for example, dispersion and flocking, when robots interact with others by applying an acute-angle test-based interaction rule. Fan et al. [26] proposed bounded control for preserving the connectivity of multiagent systems using the constraint function approach. Cheng et al. [27] designed an event-triggered optimal nonlinear-system control method on the basis of a state observer and neural network. Wang et al. [28] designed a DSC strategy using a sliding-mode differential and constructed a tan-type barrier Lyapunov function to handle the issue of full state constraints. Yoo [29] presented a recursive control design strategy for lower triangular cyber–physical systems with unknown nonlinear dynamics and unmatched time delays to the control input. Ma et al. [30] first introduced the DSC technique to the strict feedback nonlinear FOSs. Yang et al. [31] extended the DSC approach to nonstrict feedback nonlinear FOSs. However, the DSC method ignores the compensation error caused by the filters; thus, the control effect deteriorates.

The command-filter method was first proposed in [32]. The core idea of a command filter is that a filter is designed, such that its output output can track the derivative of reference signals; then, error compensation signals are constructed to weaken the impact of filtering errors on control quality. Numerous scholars have solved problems such as observer design and output feedback [33], switching system control with output constraints [34], and unknown system control direction [35] on the basis of the command filter. With increasing attention being paid to fractional-order backstepping control, research on the corresponding improved algorithms has emerged like mushrooms after rain. Wang et al. [36] proposed adaptive fault-tolerant control for switched nonlinear systems based on the command-filter technique. Wang et al. [37] designed command-filter-based adaptive neural control for nonstrict-feedback nonlinear systems with multiple actuator constraints. Li [38] first investigated the command-filter adaptive asymptotic tracking control of uncertain nonlinear systems with both time-varying parameters and uncertain disturbances, and a novel quadratic Lyapunov function by incorporating the lower bounds of control gains was proposed that guaranteed that the tracking error asymptotically converges to zero. Liu et al. [39] proposed a command-filter backstepping control scheme for fractional-order nonlinear systems with actuator faults. You et al. [40] presented a finite-time fractional-order command-filtered implementation strategy for the backstepping approach that was further applied to uncertain fractional-order nonlinear systems. The published works focused on the command-filter design of only commensurate FOSs. The proposed approach for commensurate FOSs was based on the direct Lyapunov method and requires the fractional-order derivatives of the Lyapunov functions. Therefore, it cannot be applied to incommensurate FOSs. To the best of the authors’ knowledge, there are few works on command-filter-based control schemes for incommensurate FOSs.

Motivated by the above discussions, we developed an adaptive fuzzy ETC strategy for nonstrict feedback incommensurate nonlinear FOSs with external disturbances. The main contributions of the proposed approach are summarized as follows.

1.An adaptive backstepping recursive algorithm was developed for strict feedback incommensurate nonlinear FOSs, and the stability of closed-loop systems is analyzed with the indirect Lyapunov method. Different from the results in [41,42], we considered a more general kind of systems, and the approach presented in this work has more abundant applications in practice [43,44,45]. Accordingly, it introduces challenges in the analysis and synthesis of controller design.2.The command-filter control scheme was first proposed for incommensurate FOSs to address the dimension explosion issue, and a fractional-order filter was designed. The fuzzy approximation errors were estimated with adaptive update laws. In contrast to the works in [20,21,22], the information of the high-order differentials of reference signals is not essential. In addition, closed-loop control performance can be guaranteed.3.The unknown control coefficient is considered in this paper, and the parameter update law was constructed to estimate the control coefficient. Due to the sign function introduced into the controller, the chattering phenomenon occurs. The hyperbolic tangent function was utilized to replace the sign function, proving that the stability of the closed-loop system is still guaranteed.

The remainder of this article is organized as follows. Section 2 gives the system descriptions and some useful preliminaries. Section 3 presents the synthesis process of the adaptive fuzzy command-filter control for incommensurate nonlinear FOSs. Section 4 presents the simulation results to demonstrate the effectiveness of the proposed method. Lastly, Section 5 concludes this work.

## 2. Problem Statement and Preliminaries

**Definition 1** ([46])**.**
*The Caputo fractional-order derivative with zero initial time is defined as follows:*
(1)$$D^{\alpha }F\left(s\right)=\frac{1}{\Gamma (k-\alpha )}\begin{array}{c} \int_{0}^{s}{\left(s-\tau \right)}^{k-\alpha -1}F^{\left(k\right)}\left(\tau \right)d\tau \end{array},$$*where k−1<α≤k, k∈N, F(s) is a continuously differentiable function, $F^{\left(k\right)}\left(s\right)$ denotes the traditional k-th differential order of F(s), and $\Gamma \left(\alpha \right)=\int_{0}^{\infty }\tau^{\alpha -1}e^{-\tau }d\tau$ is the gamma function.*

**Definition 2** ([46])**.**
*The fractional-order integral is defined as follows:*
(2)$$I^{\alpha }F\left(s\right)=\frac{1}{\Gamma (\alpha )}\begin{array}{c} \int_{0}^{s}{\left(s-\tau \right)}^{\alpha -1}F\left(\tau \right)d\tau \end{array},$$*where α>0.*

Before proceeding, the following useful lemmas are introduced.

**Lemma 1** ([47])**.**
*The differential equation $D^{\alpha }X\left(t\right)=p\left(t\right)$ with α∈(0,1), X(t)∈R and p(t)∈R can be transformed into the following continuous frequency distributed model (FDM):*
(3)$$\begin{array}{c} \left\{\begin{array}{c} \frac{\partial h(\sigma ,t)}{\partial t}=-\sigma h\left(\sigma ,t\right)+p\left(t\right),\;\\ X\left(t\right)=\int_{0}^{\infty }\zeta_{\alpha }\left(\sigma \right)h\left(\sigma ,t\right)d\sigma , \end{array}\right. \end{array}$$*where $\zeta_{\alpha }\left(\sigma \right)=\frac{\sin(\alpha \pi )}{\sigma^{\alpha }\pi }$.*

**Lemma 2** ([40])**.**
*Let v(t) be a signal satisfying |υ(t)|≤ψ and $|D^{\alpha }\upsilon \left(t\right)|\leq \vartheta$. For ∀t≥0 and r>0, one has*
(4)$$D^{\alpha }\omega \left(t\right)=-r\left(\omega \left(t\right)-\upsilon \left(t\right)\right),$$*where ω(0)=υ(0). There exists a constant d>0, such that $\left|\omega \left(t\right)-\upsilon \left(t\right)\right|\leq \frac{3\vartheta d}{r}$.*

**Remark 1.** 
*The above lemma indicates that selecting a sufficiently large filter coefficient r can ensure that the filtering error is small enough. Then, it is not necessary to consider filtering error signals in the design process of Lyapunov functions.*


### 2.1. System Descriptions

Consider the following uncertain nonlinear FOSs:(5)$$\begin{array}{c} \left\{\begin{array}{cc} D^{\alpha_{i}}x_{i} & =x_{i+1}+f_{i}\left({\underline{x}}_{i}\right),\;i=1,2,\cdots ,n-1,\\ D^{\alpha_{n}}x_{n} & =bu+f_{n}\left({\underline{x}}_{n}\right),\\ y & =x_{1}, \end{array}\right. \end{array}$$
where $x={\left[x_{1},x_{2},\ldots ,x_{n}\right]}^{T}\in R^{n}$ denotes the pseudostate vector, u∈R and y∈R represent the system input and output variable, respectively. $\alpha_{i}$ is the system incommensurate fractional order that satisfies that $0<\alpha_{i}<1$ for i∈N≜{1,2,…,n}, ${\underline{x}}_{i}={\left[x_{1},x_{2},\cdots ,x_{i}\right]}^{T}\in R^{i}$, *b* is the control coefficient, and $f_{i}\left(x\right)\in R$ is an unknown smooth nonlinear function.

**Control objective**: considering uncertain nonlinear FOSs (Equation 5) to design a command-filter-based adaptive fuzzy dynamic surface control approach, such that all the signals in closed-loop system are uniformly ultimately bounded, and the tracking error asymptotically converges to the equilibrium point.

**Remark 2.** 
*In most existing work, the command-filter-based dynamic surface control approach was proposed for commensurate fractional-order systems. The direct Lyapunov method is utilized to analyze the stability of closed-loop systems. It is required to take the derivative of the Lyapunov function; thus, these approaches are not applicable to incommensurate fractional-order systems. The indirect Lyapunov method is established on the basis of the frequency distributed model, and real states have an infinite dimensional property. If the Lyapunov function satisfies $\dot{V}\leq 0$, real states can converge to zero asymptotically. Then, the pseudostates converge to zero asymptotically, while, because of the existence of filter errors, condition $\dot{V}\leq 0$ cannot be obtained. To sum up, the significant difficulty is in how to analyze the stability of pseudostates when the command-filter-based method is applied to incommensurate fractional-order systems.*


Some common assumptions are as follows.

**Assumption 1.** 
*Reference signal $y_{r}$ and $D^{\alpha_{i}}y_{r}$ are available and bounded for all i∈N.*


**Assumption 2.** 
*The value of control coefficient b is unknown, while the sign of b is known.*


**Remark 3.** 
*In [20,21,22], it was required that the reference signal and its first $\sum_{i=1}^{j}\alpha_{i}$-th order derivatives were piecewise continuous and bounded, j∈N. The restrictive assumption was removed in this article; thus, the dimension explosion issue is tackled, and the computational burden is mitigated.*


**Remark 4.** 
*Assumption 1 is a common assumption for the backstepping control of incommensurate fractional-order systems; see [20,21,22]. $y_{d}$ must be bounded; otherwise, the system output cannot obviously track the reference signal. The fractional-order derivative of a reference signal is used in the design process of virtual control variables. $D^{\alpha_{i}}y_{d}$ need to be bounded and available to guarantee that the dynamic surface control method can be utilized for incommensurate fractional-order systems.*


### 2.2. Fuzzy Logic Systems

Consider the following inference rules:

$$\begin{array}{c} R^{l}:\;\mathrm{IF}\;x_{1}\;\mathrm{is}\;G_{1}^{l}\;\mathrm{AND}\ldots \mathrm{AND}\;x_{p}\;\mathrm{is}\;G_{p}^{l};\\ \mathrm{THEN}\;y\;\mathrm{is}\;y_{f}^{l}\;\left(l=1,2,\ldots ,N\right), \end{array}$$where $G_{i}^{l}$ is a fuzzy set, i∈N, $y_{f}^{l}$ is the crisp output of *l*th rule, and *N* is the total number of rules.

Through the fuzzifier, product inference, and defuzzifier, the fuzzy logic system (FLS) can be inferred as follows:(6)$$y\left(x\right)=\frac{\sum_{l=1}^{N}\mu_{l}\left(x\right)y_{f}^{l}}{\sum_{l=1}^{N}\mu_{l}\left(x\right)},$$
where $\mu_{l}\left(x\right)=\prod_{i=1}^{n}\mu_{G_{i}^{l}}\left(x_{i}\right)$, $\mu_{G_{i}^{l}}\left(x_{i}\right)$ is the membership function. By introducing the fuzzy basis functions, the FLS (Equation 6) can be expressed as follows:(7)$$y\left(x\right)=\Theta^{T}\xi \left(x\right),$$
where $\xi \left(x\right)=[\xi_{1}\left(x\right),\xi_{2}\left(x\right),\ldots ,\xi_{N}\left(x\right)]$, $\Theta =[y_{f}^{1},y_{f}^{2},\ldots ,y_{f}^{N}]$, and
(8)$$\xi_{l}\left(x\right)=\frac{\mu_{l}\left(x\right)}{\sum_{i=1}^{N}\mu_{i}\left(x\right)},\;l=1,2,\ldots ,N.$$

**Lemma 3** ([48])**.**
*For a continuous function f(x) defined on a compact set Ω, there exists a FLS (Equation 7), such that*
(9)$$\underset{x\in \Omega }{\sup}\left|f\left(x\right)-\Theta^{T}\xi \left(x\right)\right|\leq \epsilon .$$

According to Lemma 3, unknown nonlinear function $f_{i}\left({\underline{x}}_{i}\right)$ can be expressed as follows:(10)$$f_{i}\left({\underline{x}}_{i}\right)=\Theta_{i}^{T}\xi_{i}\left({\underline{x}}_{i}\right)+\delta_{i},$$
where $\delta_{i}$ is the bounded approximation error satisfying $|\delta_{i}\left({\underline{x}}_{i}\right)|\leq D_{i}$, $D_{i}$ is the upper bound of the approximation error, and
(11)$$\Theta_{i}=\underset{{\hat{\Theta }}_{i}\in \Omega_{i}}{\arg\min}\left\{\underset{\underline{x}\in X_{i}}{\sup}\left|f_{i}\left({\underline{x}}_{i}\right)-{\hat{\Theta }}_{i}^{T}\xi_{i}\left({\underline{x}}_{i}\right)\right|\right\},$$
where ${\hat{\Theta }}_{i}$ is the estimation of $\Theta_{i}$, and $\Omega_{i}$ and $X_{i}$ are compact sets for ${\hat{\Theta }}_{i}$ and ${\underline{x}}_{i}$, respectively.

## 3. Command-Filter Control Scheme Design

Before the backstepping procedure, some signals are defined as follows:(12)$$\begin{array}{c} \epsilon_{1}=y-y_{r},\;\epsilon_{i}=x_{i}-\omega_{i},\;z_{j}=\epsilon_{j}-s_{j}, \end{array}$$
where i=2,3,…,n, j=1,2,…,n, $\epsilon_{i}$ and $z_{j}$ are error surfaces, $s_{j}$ are compensation signals, $\upsilon_{i-1}$ denote virtual controllers, and $\omega_{i}$ denotes the filtered virtual signals.

The virtual controller is defined as follows:(13)$$\begin{array}{c} \left\{\begin{array}{cc} \upsilon_{1}= & -c_{1}\epsilon_{1}+D^{\alpha_{1}}y_{r}-{\hat{\Theta }}_{1}^{T}\xi_{1}\left({\underline{x}}_{1}\right)-\mathrm{sgn}\left(z_{1}\right){\hat{D}}_{1},\\ \upsilon_{i}= & -c_{i}\epsilon_{i}-\epsilon_{i-1}+D^{\alpha_{i}}\omega_{i}-{\hat{\Theta }}_{i}^{T}\xi_{i}\left({\underline{x}}_{i}\right)-\mathrm{sgn}\left(z_{i}\right){\hat{D}}_{i},\;i=2,3,\cdots ,n-1. \end{array}\right. \end{array}$$

The fractional-order filter is designed as follows:(14)$$\begin{array}{c} D^{\alpha_{i}}\omega_{i}=r_{i}\upsilon_{i-1}-r_{i}\omega_{i},\;i=2,3,\cdots ,n. \end{array}$$

The compensation signal is constructed as follows:(15)$$\begin{array}{c} \left\{\begin{array}{cc} D^{\alpha_{1}}s_{1} & =-c_{1}s_{1}+s_{2}+\omega_{2}-\upsilon_{1},\\ D^{\alpha_{i}}s_{i} & =-c_{i}s_{i}-s_{i-1}+s_{i+1}+\omega_{i+1}-\upsilon_{i},\;i=2,3,\cdots ,n-1,\\ D^{\alpha_{n}}s_{n} & =-c_{n}s_{n}-s_{n-1}. \end{array}\right. \end{array}$$

The parameter update laws are designed as follows:(16)$$\begin{array}{c} \left\{\begin{array}{cc} D^{\beta_{i}}{\hat{\Theta }}_{i} & =\pi_{i}z_{i}\xi_{i}\left({\underline{x}}_{i}\right),\;i=1,2,\cdots ,n,\\ D^{\gamma_{i}}{\hat{D}}_{i} & =\lambda_{i}\left|z_{i}\right|,\;i=1,2,\cdots ,n,\\ D^{\alpha }\hat{\varphi } & =-\tau \mathrm{sgn}\left(b\right)z_{n}\upsilon_{n}. \end{array}\right. \end{array}$$

The actual control input was chosen as follows:(17)$$\begin{array}{c} \left\{\begin{array}{cc} u= & \;\hat{\varphi }\upsilon_{n},\\ \upsilon_{n}= & -c_{n}\epsilon_{n}-\epsilon_{n-1}+D^{\alpha_{n}}\omega_{n}-{\hat{\Theta }}_{n}^{T}\xi_{n}\left(x\right)-\mathrm{sgn}\left(z_{n}\right){\hat{D}}_{n}, \end{array}\right. \end{array}$$
where $0<\alpha ,\beta_{i},\gamma_{i}<1$ are the fractional order of update laws, define $\varphi =\frac{1}{b}$, ${\hat{\Theta }}_{i}$, ${\hat{D}}_{i}$ and ${\hat{\varphi }}_{i}$ are the estimation of parameters $\Theta_{i}$, $D_{i}$, $\varphi_{i}$, respectively, $c_{i}$, $\pi_{i}$, $\lambda_{i}$, τ and $r_{j}$ are positive designed parameters, i=1,2,⋯,n, j=2,3,⋯,n.

It follows from (Equation 10), (Equation 12), and (Equation 15) that
(18)$$\begin{array}{c} \left\{\begin{array}{cc} D^{\alpha_{1}}z_{1}= & z_{2}+f_{1}\left({\underline{x}}_{1}\right)-D^{\alpha_{1}}y_{r}+c_{1}s_{1}+\upsilon_{1},\\ D^{\alpha_{i}}z_{i}= & z_{i+1}+f_{i}\left({\underline{x}}_{i}\right)-D^{\alpha_{i}}\omega_{i}+c_{i}s_{i}+s_{i-1}+\upsilon_{i},\;i=2,3,\cdots ,n-1,\\ D^{\alpha_{n}}z_{n}= & bu+f_{n}\left({\underline{x}}_{n}\right)-D^{\alpha_{n}}\omega_{n}+c_{n}s_{n}+s_{n-1}. \end{array}\right. \end{array}$$

**Step 1.** On the basis of Lemma 1, the related FDM of (Equation 18) can be obtained as follows:(19)$$\begin{array}{c} \left\{\begin{array}{cc} \frac{\partial h_{z_{1}}\left(\sigma ,t\right)}{\partial t}= & -\sigma h_{z_{1}}\left(\sigma ,t\right)+z_{2}+f_{1}\left({\underline{x}}_{1}\right)-D^{\alpha_{1}}y_{r}+c_{1}s_{1}+\upsilon_{1},\\ z_{1}= & \int_{0}^{\infty }\zeta_{\alpha_{1}}\left(\sigma \right)h_{z_{1}}\left(\sigma ,t\right)d\sigma , \end{array}\right. \end{array}$$
where $\zeta_{\alpha_{1}}\left(\sigma \right)=\frac{\sin(\alpha_{1}\pi )}{\sigma^{\alpha_{1}}\pi }$.

Define the parameter estimation errors as ${\tilde{\Theta }}_{1}=\Theta_{1}-{\hat{\Theta }}_{1}$ and ${\tilde{D}}_{1}=D_{1}-{\hat{D}}_{1}$; then, we have
(20)$$\left\{\begin{array}{cc} \frac{\partial h_{\Theta_{1}}\left(\sigma ,t\right)}{\partial t}= & -\sigma h_{\Theta_{1}}\left(\sigma ,t\right)-D^{\beta_{1}}{\hat{\Theta }}_{1}\;\\ {\tilde{\Theta }}_{1}= & \int_{0}^{\infty }\zeta_{\beta_{1}}\left(\sigma \right)h_{\Theta_{1}}\left(\sigma ,t\right)d\sigma ,\;\\ \frac{\partial h_{D_{1}}\left(\sigma ,t\right)}{\partial t}= & -\sigma h_{D_{1}}\left(\sigma ,t\right)-D^{\gamma_{1}}{\hat{D}}_{1},\;\\ {\tilde{D}}_{1}= & \int_{0}^{\infty }\zeta_{\gamma_{1}}\left(\sigma \right)h_{D_{1}}\left(\sigma ,t\right)d\sigma , \end{array}\right.$$where $\zeta_{\beta_{1}}\left(\sigma \right)=\frac{\sin(\beta_{1}\pi )}{\sigma^{\beta_{1}}\pi }$ and $\zeta_{\gamma_{1}}\left(\sigma \right)=\frac{\sin(\gamma_{1}\pi )}{\sigma^{\gamma_{1}}\pi }$.

Consider the following Lyapunov function:(21)$$\begin{array}{c} \begin{array}{cc} V_{1}= & \frac{1}{2\pi_{1}}\int_{0}^{\infty }\zeta_{\beta_{1}}\left(\sigma \right)h_{\Theta_{1}}^{T}\left(\sigma ,t\right)h_{\Theta_{1}}\left(\sigma ,t\right)d\sigma \\ & +\frac{1}{2\lambda_{1}}\int_{0}^{\infty }\zeta_{\gamma_{1}}\left(\sigma \right)h_{D_{1}}^{2}\left(\sigma ,t\right)d\sigma +\frac{1}{2}\int_{0}^{\infty }\zeta_{\alpha_{1}}\left(\sigma \right)h_{z_{1}}^{2}\left(\sigma ,t\right)d\sigma . \end{array} \end{array}$$

By substituting Virtual Control Law (Equation 13) into the above inequality and taking the derivative of $V_{1}$, one can obtain that
(22)$$\begin{array}{c} \begin{array}{cc} {\dot{V}}_{1}= & -\frac{1}{\pi_{1}}\int_{0}^{\infty }\sigma \zeta_{\beta_{1}}\left(\sigma \right)h_{\Theta_{1}}^{T}\left(\sigma ,t\right)h_{\Theta_{1}}\left(\sigma ,t\right)d\sigma -\frac{1}{\pi_{1}}{\tilde{\Theta }}_{1}^{T}D^{\beta_{1}}{\hat{\Theta }}_{1}\\ & -\frac{1}{\lambda_{1}}\int_{0}^{\infty }\sigma \zeta_{\gamma_{1}}\left(\sigma \right)h_{D_{1}}^{2}\left(\sigma ,t\right)d\sigma -\int_{0}^{\infty }\sigma \zeta_{\alpha_{1}}\left(\sigma \right)h_{z_{1}}^{2}\left(\sigma ,t\right)d\sigma \\ & -\frac{1}{\lambda_{1}}{\tilde{D}}_{1}D^{\gamma_{1}}{\hat{D}}_{1}+z_{1}z_{2}+c_{1}z_{1}\left(s_{1}-\epsilon_{1}\right)+z_{1}\left[f_{1}\left({\underline{x}}_{1}\right)-{\hat{\Theta }}_{1}^{T}\xi_{1}\left({\underline{x}}_{1}\right)\right]-\left|z_{1}\right|{\hat{D}}_{1}. \end{array} \end{array}$$

Consider the following fact:(23)$$\begin{array}{c} z_{1}\left[f_{1}\left({\underline{x}}_{1}\right)-{\hat{\Theta }}_{1}^{T}\xi_{1}\left({\underline{x}}_{1}\right)\right]\leq \left|z_{1}\right|D_{1}+z_{1}{\tilde{\Theta }}_{1}^{T}\xi_{1}\left({\underline{x}}_{1}\right). \end{array}$$

Substituting (Equation 23) into (Equation 22), we have
(24)$$\begin{array}{c} \begin{array}{cc} {\dot{V}}_{1}\leq & -\frac{1}{\pi_{1}}\int_{0}^{\infty }\sigma \zeta_{\beta_{1}}\left(\sigma \right)h_{\Theta_{1}}^{T}\left(\sigma ,t\right)h_{\Theta_{1}}\left(\sigma ,t\right)d\sigma -\frac{1}{\lambda_{1}}\int_{0}^{\infty }\sigma \zeta_{\gamma_{1}}\left(\sigma \right)h_{D_{1}}^{2}\left(\sigma ,t\right)d\sigma \\ & -\int_{0}^{\infty }\sigma \zeta_{\alpha_{1}}\left(\sigma \right)h_{z_{1}}^{2}\left(\sigma ,t\right)d\sigma -c_{1}z_{1}^{2}+z_{1}z_{2}+{\tilde{\Theta }}_{1}^{T}\left[z_{1}\xi_{1}\left({\underline{x}}_{1}\right)-\frac{1}{\pi_{1}}D^{\beta_{1}}{\hat{\Theta }}_{1}\right]\\ & +{\tilde{D}}_{1}\left(\right|z_{1}\left|-\frac{1}{\lambda_{1}}D^{\gamma_{1}}{\hat{D}}_{1}\right). \end{array} \end{array}$$

Bearing (Equation 16) in mind, we obtain that
(25)$$\begin{array}{c} \begin{array}{cc} {\dot{V}}_{1}\leq & -\frac{1}{\pi_{1}}\int_{0}^{\infty }\sigma \zeta_{\beta_{1}}\left(\sigma \right)h_{\Theta_{1}}^{T}\left(\sigma ,t\right)h_{\Theta_{1}}\left(\sigma ,t\right)d\sigma -\frac{1}{\lambda_{1}}\int_{0}^{\infty }\sigma \zeta_{\gamma_{1}}\left(\sigma \right)h_{D_{1}}^{2}\left(\sigma ,t\right)d\sigma \\ & -\int_{0}^{\infty }\sigma \zeta_{\alpha_{1}}\left(\sigma \right)h_{z_{1}}^{2}\left(\sigma ,t\right)d\sigma -c_{1}z_{1}^{2}+z_{1}z_{2}. \end{array} \end{array}$$

**Step** i(i=2,…,n−1). It follows from (Equation 18) that the FDM of $z_{i}$ can be expressed as follows:(26)$$\begin{array}{c} \left\{\begin{array}{cc} \frac{\partial h_{z_{i}}\left(\sigma ,t\right)}{\partial t}= & -\sigma h_{z_{i}}\left(\sigma ,t\right)+z_{i+1}+f_{i}\left(x\right)-D^{\alpha_{i}}\omega_{i}+c_{i}s_{i}+s_{i-1}+\upsilon_{i},\\ z_{i}= & \int_{0}^{\infty }\zeta_{\alpha_{i}}\left(\sigma \right)h_{z_{i}}\left(\sigma ,t\right)d\sigma , \end{array}\right. \end{array}$$
where $\zeta_{\alpha_{i}}\left(\sigma \right)=\frac{\sin(\alpha_{i}\pi )}{\sigma^{\alpha_{i}}\pi }$.

Select the Lyapunov function as follows:(27)$$\begin{array}{c} \begin{array}{cc} V_{i}= & V_{i-1}+\frac{1}{2\pi_{i}}\int_{0}^{\infty }\zeta_{\beta_{i}}\left(\sigma \right)h_{\Theta_{i}}^{T}\left(\sigma ,t\right)h_{\Theta_{i}}\left(\sigma ,t\right)d\sigma \\ & +\frac{1}{2\lambda_{i}}\int_{0}^{\infty }\zeta_{\gamma_{i}}\left(\sigma \right)h_{D_{i}}^{2}\left(\sigma ,t\right)d\sigma +\frac{1}{2}\int_{0}^{\infty }\zeta_{\alpha_{i}}\left(\sigma \right)h_{z_{i}}^{2}\left(\sigma ,t\right)d\sigma . \end{array} \end{array}$$

Taking the derivative of $V_{i}$, we obtain that
(28)$$\begin{array}{c} \begin{array}{cc} {\dot{V}}_{i}= & {\dot{V}}_{i-1}-\frac{1}{\pi_{i}}\int_{0}^{\infty }\sigma \zeta_{\beta_{i}}\left(\sigma \right)h_{\Theta_{i}}^{T}\left(\sigma ,t\right)h_{\Theta_{i}}\left(\sigma ,t\right)d\sigma -\frac{1}{\pi_{i}}{\tilde{\Theta }}_{i}^{T}D^{\beta_{i}}{\hat{\Theta }}_{i}\\ & -\frac{1}{\lambda_{i}}\int_{0}^{\infty }\sigma \zeta_{\gamma_{i}}\left(\sigma \right)h_{D_{i}}^{2}\left(\sigma ,t\right)d\sigma -\frac{1}{\lambda_{i}}{\tilde{D}}_{i}D^{\gamma_{i}}{\hat{D}}_{i}\\ & -\int_{0}^{\infty }\sigma \zeta_{\alpha_{i}}\left(\sigma \right)h_{z_{i}}^{2}\left(\sigma ,t\right)d\sigma +z_{i}\left[z_{i+1}+f_{i}\left({\underline{x}}_{i}\right)-D^{\alpha_{i}}\omega_{i}+c_{i}s_{i}+s_{i-1}+\upsilon_{i}\right]. \end{array} \end{array}$$

Substituting Virtual Control Law (Equation 13) into the above inequality, one has
(29)$$\begin{array}{c} \begin{array}{cc} {\dot{V}}_{i}= & {\dot{V}}_{i-1}-\frac{1}{\pi_{i}}\int_{0}^{\infty }\sigma \zeta_{\beta_{i}}\left(\sigma \right)h_{\Theta_{i}}^{T}\left(\sigma ,t\right)h_{\Theta_{i}}\left(\sigma ,t\right)d\sigma -\frac{1}{\lambda_{i}}\int_{0}^{\infty }\sigma \zeta_{\gamma_{i}}\left(\sigma \right)h_{D_{i}}^{2}\left(\sigma ,t\right)d\sigma \\ & -\int_{0}^{\infty }\sigma \zeta_{\alpha_{i}}\left(\sigma \right)h_{z_{i}}^{2}\left(\sigma ,t\right)d\sigma -\frac{1}{\pi_{i}}{\tilde{\Theta }}_{i}^{T}D^{\beta_{i}}{\hat{\Theta }}_{i}-\frac{1}{\lambda_{i}}{\tilde{D}}_{i}D^{\gamma_{i}}{\hat{D}}_{i}\\ & -c_{i}z_{i}^{2}+z_{i}z_{i+1}-z_{i}z_{i-1}+z_{i}\left[f_{i}\left({\underline{x}}_{i}\right)-{\hat{\Theta }}_{i}^{T}\xi_{i}\left({\underline{x}}_{i}\right)\right]-\left|z_{i}\right|{\hat{D}}_{i}\\ \leq & {\dot{V}}_{i-1}-\frac{1}{\pi_{i}}\int_{0}^{\infty }\sigma \zeta_{\beta_{i}}\left(\sigma \right)h_{\Theta_{i}}^{T}\left(\sigma ,t\right)h_{\Theta_{i}}\left(\sigma ,t\right)d\sigma -\frac{1}{\lambda_{i}}\int_{0}^{\infty }\sigma \zeta_{\gamma_{i}}\left(\sigma \right)h_{D_{i}}^{2}\left(\sigma ,t\right)d\sigma \\ & -\int_{0}^{\infty }\sigma \zeta_{\alpha_{i}}\left(\sigma \right)h_{z_{i}}^{2}\left(\sigma ,t\right)d\sigma -c_{i}z_{i}^{2}+z_{i}z_{i+1}-z_{i}z_{i-1}+{\tilde{\Theta }}_{i}^{T}\left[z_{i}\xi_{i}\left({\underline{x}}_{i}\right)-\frac{1}{\pi_{i}}D^{\beta_{i}}{\hat{\Theta }}_{i}\right]\\ & +{\tilde{D}}_{i}\left(\right|z_{i}\left|-\frac{1}{\lambda_{i}}D^{\gamma_{i}}{\hat{D}}_{i}\right). \end{array} \end{array}$$

Bearing (Equation 16) in mind, it can be obtained that
(30)$$\begin{array}{c} \begin{array}{cc} {\dot{V}}_{i}\leq & {\dot{V}}_{i-1}-\frac{1}{\pi_{i}}\int_{0}^{\infty }\sigma \zeta_{\beta_{i}}\left(\sigma \right)h_{\Theta_{i}}^{T}\left(\sigma ,t\right)h_{\Theta_{i}}\left(\sigma ,t\right)d\sigma -\frac{1}{\lambda_{i}}\int_{0}^{\infty }\sigma \zeta_{\gamma_{i}}\left(\sigma \right)h_{D_{i}}^{2}\left(\sigma ,t\right)d\sigma \\ & -\int_{0}^{\infty }\sigma \zeta_{\alpha_{i}}\left(\sigma \right)h_{z_{i}}^{2}\left(\sigma ,t\right)d\sigma -c_{i}z_{i}^{2}+z_{i}z_{i+1}-z_{i}z_{i-1}. \end{array} \end{array}$$

The following theorem summarizes the main results of the controller design.

**Theorem 1.** 
*Considering that System (Equation 5) satisfies Assumptions 1 and 2, the virtual controllers were designed as in (Equation 13), the fractional-order filter was designed as in (Equation 14), the actual control signal was chosen as in (Equation 17), and the update laws were selected as in (Equation 16). Then, the presented method guaranteed that all signals of closed-loop System (Equation 5) were bounded, and the tracking error could asymptotically converge to the equilibrium point.*


**Proof.** **Step** *n*. By defining parameter estimation error $\tilde{\varphi }=\varphi -\hat{\varphi }$, the fractional-order derivative of $\tilde{\varphi }$ can be expressed as follows:
(31)$$\begin{array}{c} \left\{\begin{array}{cc} \frac{\partial h_{\varphi }\left(\sigma ,t\right)}{\partial t}= & -\sigma h_{\varphi }\left(\sigma ,t\right)-D^{\alpha }\hat{\varphi },\\ \tilde{\varphi }= & \int_{0}^{\infty }\zeta_{\alpha }\left(\sigma \right)h_{\varphi }\left(\sigma ,t\right)d\sigma . \end{array}\right. \end{array}$$
where $\zeta_{\alpha }\left(\sigma \right)=\frac{\sin(\alpha \pi )}{\sigma^{\alpha }\pi }$.Choose the Lyapunov function as follows:
(32)$$\begin{array}{c} \begin{array}{cc} V_{n}= & V_{n-1}+\frac{1}{2\pi_{n}}\int_{0}^{\infty }\zeta_{\beta_{n}}\left(\sigma \right)h_{\Theta_{n}}^{T}\left(\sigma ,t\right)h_{\Theta_{n}}\left(\sigma ,t\right)d\sigma +\frac{\left|b\right|}{2\tau }\int_{0}^{\infty }\zeta_{\alpha }\left(\sigma \right)h_{\varphi }^{2}\left(\sigma ,t\right)d\sigma \\ & +\frac{1}{2\lambda_{n}}\int_{0}^{\infty }\zeta_{\gamma_{n}}\left(\sigma \right)h_{D_{n}}^{2}\left(\sigma ,t\right)d\sigma +\frac{1}{2}\int_{0}^{\infty }\zeta_{\alpha_{n}}\left(\sigma \right)h_{z_{n}}^{2}\left(\sigma ,t\right)d\sigma . \end{array} \end{array}$$By substituting Virtual Control Law (Equation 13) into the above inequality and taking the derivative of $V_{1}$, one has
(33)$$\begin{array}{c} \begin{array}{cc} {\dot{V}}_{n}= & {\dot{V}}_{n-1}-\frac{1}{\pi_{n}}\int_{0}^{\infty }\sigma \zeta_{\beta_{n}}\left(\sigma \right)h_{\Theta_{n}}^{T}\left(\sigma ,t\right)h_{\Theta_{n}}\left(\sigma ,t\right)d\sigma -\frac{1}{\lambda_{n}}\int_{0}^{\infty }\sigma \zeta_{\gamma_{n}}\left(\sigma \right)h_{D_{n}}^{2}\left(\sigma ,t\right)d\sigma \\ & -\int_{0}^{\infty }\sigma \zeta_{\alpha_{n}}\left(\sigma \right)h_{z_{n}}^{2}\left(\sigma ,t\right)d\sigma -\frac{\left|b\right|}{\tau }\int_{0}^{\infty }\sigma \zeta_{\alpha }\left(\sigma \right)h_{\varphi }^{2}\left(\sigma ,t\right)d\sigma -\frac{1}{\pi_{n}}{\tilde{\Theta }}_{n}^{T}D^{\beta_{n}}{\hat{\Theta }}_{n}\\ & -\frac{1}{\lambda_{n}}{\tilde{D}}_{n}D^{\gamma_{n}}{\hat{D}}_{n}+z_{n}\left[\upsilon_{n}+f_{n}\left(x\right)-D^{\alpha_{n}}\omega_{n}+c_{n}s_{n}+s_{n-1}\right]\\ & -b\tilde{\varphi }z_{n}\upsilon_{n}-\frac{\left|b\right|}{\tau }\tilde{\varphi }D^{\alpha }\hat{\varphi }. \end{array} \end{array}$$By designing the update law as in (Equation 16) and substituting it into the derivative of the Lyapunov function, we have
$$\begin{array}{c} \begin{array}{cc} {\dot{V}}_{n}\leq & {\dot{V}}_{n-1}-\frac{1}{\pi_{n}}\int_{0}^{\infty }\sigma \zeta_{\beta_{n}}\left(\sigma \right)h_{\Theta_{n}}^{T}\left(\sigma ,t\right)h_{\Theta_{n}}\left(\sigma ,t\right)d\sigma -\frac{1}{\lambda_{n}}\int_{0}^{\infty }\sigma \zeta_{\gamma_{n}}\left(\sigma \right)h_{D_{n}}^{2}\left(\sigma ,t\right)d\sigma \\ & -\int_{0}^{\infty }\sigma \zeta_{\alpha_{n}}\left(\sigma \right)h_{z_{n}}^{2}\left(\sigma ,t\right)d\sigma -\frac{\left|b\right|}{\tau_{1}}\int_{0}^{\infty }\sigma \zeta_{\alpha }\left(\sigma \right)h_{\varphi }^{2}\left(\sigma ,t\right)d\sigma -c_{n}z_{n}^{2}-z_{n}z_{n-1}\\ & +{\tilde{\Theta }}_{n}^{T}\left[z_{n}\xi_{n}\left(x\right)-\frac{1}{\pi_{n}}D^{\beta_{n}}{\hat{\Theta }}_{n}\right]+{\tilde{D}}_{n}\left(\right|z_{n}\left|-\frac{1}{\lambda_{n}}D^{\gamma_{n}}{\hat{D}}_{n}\right)-\tilde{\varphi }\left(bz_{n}\upsilon_{n}-\frac{\left|b\right|}{\tau }D^{\alpha }\hat{\varphi }\right)\\ \leq & -\sum_{j=1}^{n}\frac{1}{\pi_{j}}\int_{0}^{\infty }\sigma \zeta_{\beta_{j}}\left(\sigma \right)h_{\Theta_{j}}^{T}\left(\sigma ,t\right)h_{\Theta_{j}}\left(\sigma ,t\right)d\sigma -\sum_{j=1}^{n}\frac{1}{\lambda_{j}}\int_{0}^{\infty }\sigma \zeta_{\gamma_{j}}\left(\sigma \right)h_{D_{j}}^{2}\left(\sigma ,t\right)d\sigma \\ & -\sum_{j=1}^{n}\int_{0}^{\infty }\sigma \zeta_{\alpha_{j}}\left(\sigma \right)h_{z_{j}}^{2}\left(\sigma ,t\right)d\sigma -\frac{\left|b\right|}{\tau_{1}}\int_{0}^{\infty }\sigma \zeta_{\alpha }\left(\sigma \right)h_{\varphi }^{2}\left(\sigma ,t\right)d\sigma -\sum_{j=1}^{n}c_{j}z_{j}^{2}\\ \leq & 0. \end{array} \end{array}$$According to (Equation 35), as in t→∞, the real states of system $h_{\Theta_{i}}\left(\sigma ,t\right),h_{D_{i}}\left(\sigma ,t\right),h_{z_{i}}\left(\sigma ,t\right),$i=1,2,⋯,n and $h_{\varphi }\left(\sigma ,t\right)$ converge to zero. The pseudostates of closed-loop system $z_{i},{\tilde{\Theta }}_{i},{\tilde{D}}_{i},\;i=1,2,\cdots ,n$ and $\tilde{\varphi }$ accordingly converge to zero. □

**Remark 5.** 
*The approaches proposed in [30,31] were based on the direct Lyapunov method and require the fractional-order derivatives of Lyapunov functions. Therefore, it cannot be applied to incommensurate FOSs. The indirect Lyapunov method is utilized to handle control problems of incommensurate FOSs, and Lyapunov functions are designed on the basis of the real states of systems. In most existing works, the convergence of the real states is determined first; then, the convergence of its pseudostates is judged. In this paper, we do not discuss the complex relationship between real states and pseudostates, and we could directly conclude that the pseudostates were bounded.*


**Remark 6.** 
*According to the process of control scheme design, the following parameter adjustment conclusions could be obtained. By increasing parameter $c_{i},r_{i}$ and decreasing μ, the bound of the tracking error is lessened, but the transmission frequency is increased. Therefore, there exists a trade-off between tracking quality and the number of actuator updates.*


The sign function was introduced into the design of Virtual Control Law (Equation 13) and Actual Control Input (Equation 17). This led to the chattering performance shown in the simulation results. An improved method is proposed in the following theorem, and the chattering phenomenon is avoided.

**Theorem 2.** 
*Considering System (Equation 5) with Assumptions 1 and 2 satisfied, the virtual controllers, fractional-order filter, actual control signal, and parameter update laws were designed as follows. The virtual control law is as follows:*

(34)
$$\begin{array}{c} \left\{\begin{array}{cc} \upsilon_{1}= & -c_{1}\epsilon_{1}+D^{\alpha_{1}}y_{r}-{\hat{\Theta }}_{1}^{T}\xi_{1}\left({\underline{x}}_{1}\right)-\tanh\left(\frac{z_{1}}{\rho }\right){\hat{D}}_{1},\\ \upsilon_{i}= & -c_{i}\epsilon_{i}-\epsilon_{i-1}+D^{\alpha_{i}}\omega_{i}-{\hat{\Theta }}_{i}^{T}\xi_{i}\left({\underline{x}}_{i}\right)-\tanh\left(\frac{z_{i}}{\rho }\right){\hat{D}}_{i},\;i=2,3,\cdots ,n-1. \end{array}\right. \end{array}$$


*The fractional-order filter is as follows:*

(35)
$$\begin{array}{c} D^{\alpha_{i}}\omega_{i}=r_{i}\upsilon_{i-1}-r_{i}\omega_{i},\;i=2,3,\cdots ,n. \end{array}$$


*The parameter update law is as follows:*

(36)
$$\begin{array}{c} \left\{\begin{array}{cc} D^{\beta_{i}}{\hat{\Theta }}_{i} & =\pi_{i,1}\epsilon_{i}\xi_{i}\left({\underline{x}}_{i}\right)-\pi_{i,2}{\hat{\Theta }}_{i},\;i=1,2,\cdots ,n,\\ D^{\gamma_{i}}{\hat{D}}_{i} & =\lambda_{i,1}\left|\epsilon_{i}\right|-\lambda_{i,2}{\hat{D}}_{i},\;i=1,2,\cdots ,n,\\ D^{\alpha }\hat{\varphi } & =-\tau_{1}\mathrm{sgn}\left(b\right)\epsilon_{n}\upsilon_{n}-\tau_{2}\hat{\varphi }, \end{array}\right. \end{array}$$


*The actual control input is as follows:*

(37)
$$\begin{array}{c} \left\{\begin{array}{cc} u= & \;\hat{\varphi }\upsilon_{n},\\ \upsilon_{n}= & -c_{n}\epsilon_{n}-\epsilon_{n-1}+D^{\alpha_{n}}\omega_{n}-{\hat{\Theta }}_{n}^{T}\xi_{n}\left(x\right)-\tanh\left(\frac{z_{n}}{\rho }\right){\hat{D}}_{n}, \end{array}\right. \end{array}$$

*where $0<\alpha ,\beta_{i},\gamma_{i}<1$ are the fractional order of update laws defined as $\varphi =\frac{1}{b}$, ${\hat{\Theta }}_{i}$, ${\hat{D}}_{i}$ and ${\hat{\varphi }}_{i}$ are the estimation of parameters $\Theta_{i}$, $D_{i}$, $\varphi_{i}$, respectively, $c_{i}$, $\pi_{i}$, $\lambda_{i}$, τ, ρ and $r_{j}$ are designed positive parameters i=1,2,⋯,n, j=2,3,⋯,n. Then, the presented method guarantees that all signals of closed-loop System (Equation 5) are bounded, and the tracking error can converge to a small neighbourhood of the equilibrium point.*


**Proof. ** **Step 1.** Consider the following Lyapunov function:
(38)$$\begin{array}{c} \begin{array}{cc} V_{1}= & \frac{1}{2\pi_{1,1}}\int_{0}^{\infty }\zeta_{\beta_{1}}\left(\sigma \right)h_{\Theta_{1}}^{T}\left(\sigma ,t\right)h_{\Theta_{1}}\left(\sigma ,t\right)d\sigma \\ & +\frac{1}{2\lambda_{1,1}}\int_{0}^{\infty }\zeta_{\gamma_{1}}\left(\sigma \right)h_{D_{1}}^{2}\left(\sigma ,t\right)d\sigma +\frac{1}{2}\int_{0}^{\infty }\zeta_{\alpha_{1}}\left(\sigma \right)h_{z_{1}}^{2}\left(\sigma ,t\right)d\sigma . \end{array} \end{array}$$Substituting Virtual Control Law (Equation 35) into the above inequality and taking the derivative of $V_{1}$, one can obtain that
(39)$$\begin{array}{c} \begin{array}{cc} {\dot{V}}_{1}= & -\frac{1}{\pi_{1,1}}\int_{0}^{\infty }\sigma \zeta_{\beta_{1}}\left(\sigma \right)h_{\Theta_{1}}^{T}\left(\sigma ,t\right)h_{\Theta_{1}}\left(\sigma ,t\right)d\sigma -\frac{1}{\lambda_{1,1}}\int_{0}^{\infty }\sigma \zeta_{\gamma_{1}}\left(\sigma \right)h_{D_{1}}^{2}\left(\sigma ,t\right)d\sigma \\ & -\int_{0}^{\infty }\sigma \zeta_{\alpha_{1}}\left(\sigma \right)h_{z_{1}}^{2}\left(\sigma ,t\right)d\sigma -\frac{1}{\pi_{1,1}}{\tilde{\Theta }}_{1}^{T}D^{\beta_{1}}{\hat{\Theta }}_{1}-\frac{1}{\lambda_{1,1}}{\tilde{D}}_{1}D^{\gamma_{1}}{\hat{D}}_{1}\\ & +z_{1}z_{2}+c_{1}z_{1}\left(s_{1}-\epsilon_{1}\right)+z_{1}\left[f_{1}\left({\underline{x}}_{1}\right)-{\hat{\Theta }}_{1}^{T}\xi_{1}\left({\underline{x}}_{1}\right)\right]-z_{1}\tanh\left(\frac{z_{1}}{\rho }\right){\hat{D}}_{1}.\\ \leq & -\frac{1}{\pi_{1,1}}\int_{0}^{\infty }\sigma \zeta_{\beta_{1}}\left(\sigma \right)h_{\Theta_{1}}^{T}\left(\sigma ,t\right)h_{\Theta_{1}}\left(\sigma ,t\right)d\sigma -\frac{1}{\lambda_{1,1}}\int_{0}^{\infty }\sigma \zeta_{\gamma_{1}}\left(\sigma \right)h_{D_{1}}^{2}\left(\sigma ,t\right)d\sigma \\ & -\int_{0}^{\infty }\sigma \zeta_{\alpha_{1}}\left(\sigma \right)h_{z_{1}}^{2}\left(\sigma ,t\right)d\sigma -c_{1}z_{1}^{2}+z_{1}z_{2}+{\tilde{\Theta }}_{1}^{T}\left[z_{1}\xi_{1}\left({\underline{x}}_{1}\right)-\frac{1}{\pi_{1,1}}D^{\beta_{1}}{\hat{\Theta }}_{1}\right]\\ & +{\tilde{D}}_{1}\left(\right|z_{1}|-\frac{1}{\lambda_{1,1}}D^{\gamma_{1}}{\hat{D}}_{1})+{\hat{D}}_{1}\left[\right|z_{1}\left|-z_{1}\tanh\left(\frac{z_{1}}{\rho }\right)\right]. \end{array} \end{array}$$According to the literature [49], the following fact can be deduced:
(40)$$\begin{array}{c} |z_{1}|-z_{1}\tanh\left(\frac{z_{1}}{\rho }\right)\leq \rho \psi , \end{array}$$
where ψ is the solution of equation $\psi =e^{-(\psi +1)}$.Substituting (Equation 41) into (Equation 40), we have
(41)$$\begin{array}{c} \begin{array}{cc} {\dot{V}}_{1}\leq & -\frac{1}{\pi_{1,1}}\int_{0}^{\infty }\sigma \zeta_{\beta_{1}}\left(\sigma \right)h_{\Theta_{1}}^{T}\left(\sigma ,t\right)h_{\Theta_{1}}\left(\sigma ,t\right)d\sigma -\frac{1}{\lambda_{1,1}}\int_{0}^{\infty }\sigma \zeta_{\gamma_{1}}\left(\sigma \right)h_{D_{1}}^{2}\left(\sigma ,t\right)d\sigma \\ & -\int_{0}^{\infty }\sigma \zeta_{\alpha_{1}}\left(\sigma \right)h_{z_{1}}^{2}\left(\sigma ,t\right)d\sigma -c_{1}z_{1}^{2}+z_{1}z_{2}+\frac{\pi_{1,2}}{\pi_{1,1}}{\tilde{\Theta }}_{1}^{T}{\hat{\Theta }}_{1}+\frac{\lambda_{1,2}}{\lambda_{1,1}}{\tilde{D}}_{1}{\hat{D}}_{1}+\rho \psi \left|{\hat{D}}_{1}\right|\\ \leq & -\frac{1}{\pi_{1,1}}\int_{0}^{\infty }\sigma \zeta_{\beta_{1}}\left(\sigma \right)h_{\Theta_{1}}^{T}\left(\sigma ,t\right)h_{\Theta_{1}}\left(\sigma ,t\right)d\sigma -\frac{1}{\lambda_{1,1}}\int_{0}^{\infty }\sigma \zeta_{\gamma_{1}}\left(\sigma \right)h_{D_{1}}^{2}\left(\sigma ,t\right)d\sigma \\ & -\int_{0}^{\infty }\sigma \zeta_{\alpha_{1}}\left(\sigma \right)h_{z_{1}}^{2}\left(\sigma ,t\right)d\sigma -c_{1}z_{1}^{2}+z_{1}z_{2}-\frac{\pi_{1,2}}{2\pi_{1,1}}{\tilde{\Theta }}_{1}^{T}{\tilde{\Theta }}_{1}-\left(\frac{\lambda_{1,2}}{2\lambda_{1,1}}-\rho \psi \right){\tilde{D}}_{1}^{2}\\ & +\frac{\pi_{1,2}}{2\pi_{1,1}}\Theta_{1}^{T}\Theta_{1}+\left(\frac{\lambda_{1,2}}{2\lambda_{1,1}}+\rho \psi \right)D_{1}^{2}. \end{array} \end{array}$$**Step** i(i=2,…,n−1). Consider the following Lyapunov function:
(42)$$\begin{array}{c} \begin{array}{cc} V_{i}= & V_{i-1}+\frac{1}{2\pi_{i,1}}\int_{0}^{\infty }\zeta_{\beta_{i}}\left(\sigma \right)h_{\Theta_{i}}^{T}\left(\sigma ,t\right)h_{\Theta_{i}}\left(\sigma ,t\right)d\sigma \\ & +\frac{1}{2\lambda_{i,1}}\int_{0}^{\infty }\zeta_{\gamma_{i}}\left(\sigma \right)h_{D_{i}}^{2}\left(\sigma ,t\right)d\sigma +\frac{1}{2}\int_{0}^{\infty }\zeta_{\alpha_{i}}\left(\sigma \right)h_{z_{i}}^{2}\left(\sigma ,t\right)d\sigma . \end{array} \end{array}$$By substituting Virtual Control Law (Equation 35) into the above inequality and taking the derivative of $V_{i}$, one can obtain that
(43)$$\begin{array}{c} \begin{array}{cc} {\dot{V}}_{i}\leq & {\dot{V}}_{i-1}-\frac{1}{\pi_{i,1}}\int_{0}^{\infty }\sigma \zeta_{\beta_{i}}\left(\sigma \right)h_{\Theta_{i}}^{T}\left(\sigma ,t\right)h_{\Theta_{i}}\left(\sigma ,t\right)d\sigma -\frac{1}{\lambda_{i,1}}\int_{0}^{\infty }\sigma \zeta_{\gamma_{i}}\left(\sigma \right)h_{D_{i}}^{2}\left(\sigma ,t\right)d\sigma \\ & -\int_{0}^{\infty }\sigma \zeta_{\alpha_{i}}\left(\sigma \right)h_{z_{i}}^{2}\left(\sigma ,t\right)d\sigma -c_{i}z_{i}^{2}+z_{i}z_{i+1}-z_{i}z_{i-1}+{\tilde{\Theta }}_{i}^{T}\left[z_{i}\xi_{i}\left({\underline{x}}_{i}\right)-\frac{1}{\pi_{i,1}}D^{\beta_{i}}{\hat{\Theta }}_{i}\right]\\ & +{\tilde{D}}_{i}\left(\right|\epsilon_{i}|-\frac{1}{\lambda_{i,1}}D^{\gamma_{i}}{\hat{D}}_{i})+\rho \psi |{\hat{D}}_{i}|. \end{array} \end{array}$$Substituting Parameter Update Law (Equation 37) into (Equation 44), one has
(44)$$\begin{array}{c} \begin{array}{cc} {\dot{V}}_{i}\leq & {\dot{V}}_{i-1}-\frac{1}{\pi_{i,1}}\int_{0}^{\infty }\sigma \zeta_{\beta_{i}}\left(\sigma \right)h_{\Theta_{i}}^{T}\left(\sigma ,t\right)h_{\Theta_{i}}\left(\sigma ,t\right)d\sigma -\frac{1}{\lambda_{i,1}}\int_{0}^{\infty }\sigma \zeta_{\gamma_{i}}\left(\sigma \right)h_{D_{i}}^{2}\left(\sigma ,t\right)d\sigma \\ & -\int_{0}^{\infty }\sigma \zeta_{\alpha_{i}}\left(\sigma \right)h_{z_{i}}^{2}\left(\sigma ,t\right)d\sigma -c_{i}z_{i}^{2}+z_{i}z_{i+1}-z_{i}z_{i-1}-\frac{\pi_{i,2}}{2\pi_{i,1}}{\tilde{\Theta }}_{i}^{T}{\tilde{\Theta }}_{i}-\left(\frac{\lambda_{i,2}}{2\lambda_{i,1}}-\rho \psi \right){\tilde{D}}_{i}^{2}\\ & +\frac{\pi_{i,2}}{2\pi_{i,1}}\Theta_{i}^{T}\Theta_{i}+\left(\frac{\lambda_{i,2}}{2\lambda_{i,1}}+\rho \psi \right)D_{i}^{2}. \end{array} \end{array}$$**Step ***n*. Choose the following Lyapunov function:
(45)$$\begin{array}{c} \begin{array}{cc} V_{n}= & V_{n-1}+\frac{1}{2\pi_{n,1}}\int_{0}^{\infty }\zeta_{\beta_{n}}\left(\sigma \right)h_{\Theta_{n}}^{T}\left(\sigma ,t\right)h_{\Theta_{n}}\left(\sigma ,t\right)d\sigma +\frac{\left|b\right|}{2\tau_{1}}\int_{0}^{\infty }\zeta_{\alpha }\left(\sigma \right)h_{\varphi }^{2}\left(\sigma ,t\right)d\sigma \\ & +\frac{1}{2\lambda_{n,1}}\int_{0}^{\infty }\zeta_{\gamma_{n}}\left(\sigma \right)h_{D_{n}}^{2}\left(\sigma ,t\right)d\sigma +\frac{1}{2}\int_{0}^{\infty }\zeta_{\alpha_{n}}\left(\sigma \right)h_{z_{n}}^{2}\left(\sigma ,t\right)d\sigma \end{array} \end{array}$$By substituting Actual Control Input (Equation 38) into (Equation 46) and taking the derivative of $V_{n}$, we have
(46)$$\begin{array}{c} \begin{array}{cc} {\dot{V}}_{n}= & {\dot{V}}_{n-1}-\frac{1}{\pi_{n,1}}\int_{0}^{\infty }\sigma \zeta_{\beta_{n}}\left(\sigma \right)h_{\Theta_{n}}^{T}\left(\sigma ,t\right)h_{\Theta_{n}}\left(\sigma ,t\right)d\sigma -\frac{1}{\lambda_{n,1}}\int_{0}^{\infty }\sigma \zeta_{\gamma_{n}}\left(\sigma \right)h_{D_{n}}^{2}\left(\sigma ,t\right)d\sigma \\ & -\int_{0}^{\infty }\sigma \zeta_{\alpha_{n}}\left(\sigma \right)h_{z_{n}}^{2}\left(\sigma ,t\right)d\sigma -\frac{1}{\pi_{n,1}}{\tilde{\Theta }}_{n}^{T}D^{\beta_{n}}{\hat{\Theta }}_{n}-\frac{\left|b\right|}{\tau_{1}}\int_{0}^{\infty }\sigma \zeta_{\alpha }\left(\sigma \right)h_{\varphi }^{2}\left(\sigma ,t\right)d\sigma -\frac{1}{\lambda_{n,1}}{\tilde{D}}_{n}D^{\gamma_{n}}{\hat{D}}_{n}\\ & -b\tilde{\varphi }z_{n}\upsilon_{n}-\frac{\left|b\right|}{\tau_{1}}\tilde{\varphi }D^{\alpha }\hat{\varphi }+z_{n}\left[-c_{n}\epsilon_{n}-z_{n-1}-{\hat{\Theta }}_{n}^{T}\xi_{n}\left(x\right)-\tanh\left(\frac{z_{n}}{\rho }\right){\hat{D}}_{n}+f_{n}\left(x\right)\right]. \end{array} \end{array}$$Bearing Update Law (Equation 37) in mind, we can obtain that
(47)$$\begin{array}{c} \begin{array}{cc} {\dot{V}}_{n}\leq & -\sum_{j=1}^{n}\frac{1}{\pi_{j,1}}\int_{0}^{\infty }\sigma \zeta_{\beta_{j}}\left(\sigma \right)h_{\Theta_{j}}^{T}\left(\sigma ,t\right)h_{\Theta_{j}}\left(\sigma ,t\right)d\sigma -\sum_{j=1}^{n}\frac{1}{\lambda_{j,1}}\int_{0}^{\infty }\sigma \zeta_{\gamma_{j}}\left(\sigma \right)h_{D_{j}}^{2}\left(\sigma ,t\right)d\sigma \\ & -\sum_{j=1}^{n}\int_{0}^{\infty }\sigma \zeta_{\alpha_{j}}\left(\sigma \right)h_{j}^{2}\left(\sigma ,t\right)d\sigma -\sum_{j=1}^{n}c_{j}z_{j}^{2}-\frac{\left|b\right|}{\tau_{1}}\int_{0}^{\infty }\sigma \zeta_{\alpha }\left(\sigma \right)h_{\varphi }^{2}\left(\sigma ,t\right)d\sigma -\sum_{j=1}^{n}\frac{\pi_{j,2}}{2\pi_{j,1}}{\tilde{\Theta }}_{j}^{T}{\tilde{\Theta }}_{j}\\ & -\sum_{j=1}^{n}\left(\frac{\lambda_{j,2}}{2\lambda_{j,1}}-\rho \psi \right){\tilde{D}}_{j}^{2}-\frac{\left|b\right|\tau_{2}}{2\tau_{1}}{\tilde{\varphi }}^{2}+\sum_{j=1}^{n}\frac{\pi_{j,2}}{2\pi_{j,1}}\Theta_{j}^{T}\Theta_{j}+\sum_{j=1}^{n}\left(\frac{\lambda_{j,2}}{2\lambda_{j,1}}+\rho \psi \right)D_{j}^{2}+\frac{\left|b\right|\tau_{2}}{2\tau_{1}}\varphi^{2}, \end{array} \end{array}$$
where parameter $\lambda_{i,1},\lambda_{i,2}$ is selected to satisfy $\frac{\lambda_{i,2}}{\lambda_{i,1}}-\rho \psi >0,\;\left(i=1,2,\cdots ,n\right)$.Define that
(48)$$\begin{array}{c} \begin{array}{cc} C=\min & \left\{c_{1},\cdots ,c_{n},\frac{\pi_{1,2}}{2\pi_{1,1}},\cdots ,\frac{\pi_{n,2}}{2\pi_{n,1}},\frac{\lambda_{1,2}}{2\lambda_{1,1}}-\rho \psi ,\cdots ,\frac{\lambda_{n,2}}{2\lambda_{n,1}}-\rho \psi ,\frac{\left|b\right|\tau_{2}}{2\tau_{1}}\right\} \end{array} \end{array}$$
and $K=\sum_{j=1}^{n}\frac{\pi_{j,2}}{2\pi_{j,1}}\Theta_{j}^{T}\Theta_{j}+\sum_{j=1}^{n}\left(\frac{\lambda_{j,2}}{2\lambda_{j,1}}+\rho \psi \right)D_{j}^{2}+\frac{\left|b\right|\tau_{2}}{2\tau_{1}}\varphi^{2}$. It follows from (Equation 48) that
(49)$$\begin{array}{c} \begin{array}{c} {\dot{V}}_{n}\leq -C\sum_{j=1}^{n}z_{j}^{2}-C\sum_{j=1}^{n}{\tilde{\Theta }}_{j}^{T}{\tilde{\Theta }}_{j}-C\sum_{j=1}^{n}{\tilde{D}}_{j}^{2}-C{\tilde{\varphi }}^{2}+K. \end{array} \end{array}$$According to LaSalle’s invariance principle, system signals converge to the region $\left\{\left(z_{1},{\tilde{\Theta }}_{1},{\tilde{D}}_{1},\cdots ,z_{n},{\tilde{\Theta }}_{n},{\tilde{D}}_{n},\tilde{\varphi }\right)|{\dot{V}}_{n}=0\right\}$. Bearing (Equation 50) in mind, one can conclude that, as t→∞, error surfaces $z_{i}$, parameter estimation errors ${\tilde{\Theta }}_{i}$, and ${\tilde{D}}_{i}$ are bounded on compact set $E=\left\{\sum_{j=1}^{n}z_{j}^{2}+\sum_{j=1}^{n}{\tilde{\theta }}_{j}^{2}+\sum_{j=1}^{n}{\tilde{D}}_{j}^{2}+{\tilde{\varphi }}^{2}\leq \frac{K}{C}\right\}$. Thus, all the signals of closed-loop systems are bounded. □

**Remark 7.** 
*In Theorem 2, hyperbolic tangent function is utilized to replace the sign function in the controller design. The hyperbolic tangent function is an approximation of the sign function; thus, an approximation error is introduced into the closed-loop system. This leads to the tracking error not being able to asymptotically converge to zero. Therefore, the method proposed in Theorem 2 can avoid the chattering phenomenon in the control input, while steady-state performance is affected. This is a trade-off in practical application.*


## 4. Numerical Examples

**Example 1.** 
*Consider the following uncertain incommensurate nonlinear fractional-order system:*

(50)
$$\begin{array}{c} \left\{\begin{array}{cc} D^{0.5}x_{1} & =x_{2}+f_{1}\left({\underline{x}}_{1}\right),\\ D^{0.8}x_{2} & =bu+f_{2}\left({\underline{x}}_{2}\right), \end{array}\right. \end{array}$$

*where uncertain system model $f_{1}\left({\underline{x}}_{1}\right)=-0.5x_{1}^{2}$, $f_{2}\left({\underline{x}}_{2}\right)=-x_{2}+\frac{x_{2}-x_{2}^{2}}{1+x_{1}^{2}}-\sin\left(x_{1}\right)$, unknown control coefficient b=5, and the reference signal is $y_{r}\left(t\right)=0.5\left[\sin\left(2t+\frac{\pi }{4}\right)+\sin\left(4t\right)\right]$. The control parameters were adopted to be $c_{1}=c_{2}=1$, $\beta_{1}=\beta_{2}=\gamma_{1}=\gamma_{2}=0.8$, α=0.9, $\pi_{1}=\pi_{2}=\lambda_{1}=\lambda_{2}=1$, $\varphi_{1}=\varphi_{2}=\tau_{1}=\tau_{2}=1$, and $r_{2}=20$. Given the zero initial value of estimation parameters and initial state $x\left(0\right)={\left[1\;0\right]}^{T}$. In Case 1, the following simulation results could be obtained by applying the approach proposed in Theorem 1 to the FOS in (Equation 51). Figure 1 illustrates the reference signal, system output, and tracking error, Figure 2 and Figure 3 show the estimation error of the upper bound of approximation error D and unknown control coefficient b, respectively, and Figure 4 depicts the actual control input.*


The system output could track the reference signal with satisfying control performance. However, there existed an obvious chattering phenomenon in the actual control input that is not acceptable in practical application. In Case 2, the following simulation results show the effect of the improved method presented in Theorem 2. The control parameters and the initial values of system were chosen to be the same as those in the above simulation. Figure 5 illustrates the reference signal, system output, and tracking error, Figure 6 and Figure 7 show the estimation error of the upper bound of approximation error *D* and unknown control coefficient *b*, respectively, and Figure 8 depicts the actual control input.

The chattering phenomenon disappeared by utilizing the method proposed in Theorem 2. The tracking error could converge to a small neighborhood of equilibrium points. The approach in [40] was proposed for commensurate fractional-order systems via the direct Lyapunov method, which cannot be applied to this example. This reflects the progressiveness of the method proposed in this paper.

**Example 2.** 
*Consider the following uncertain incommensurate nonlinear fractional-order system:*

(51)
$$\begin{array}{c} \left\{\begin{array}{c} D^{0.8}x_{1}=x_{2}+f_{1}\left(x\right),\\ D^{0.9}x_{2}=u+f_{2}\left(x\right),\\ y=x_{1}, \end{array}\right. \end{array}$$

*where uncertain system model $f_{1}\left(x\right)=0.5\left(-x_{1}+x_{2}\right)$, $f_{2}\left(x\right)=x_{1}x_{2}$, the unknown control coefficient was chosen to be b=5, and the reference signal was chosen to be $y_{d}\left(t\right)=\sin\left(1.5t+\frac{\pi }{4}\right)$. The adopted control parameters were $c_{1}=c_{2}=c=15$, $\beta_{1}=\beta_{2}=\gamma_{1}=\gamma_{2}=G=0.8$, $\pi_{1}=\pi_{2}=\lambda_{1}=\lambda_{2}=100$, $\varphi_{1}=\varphi_{2}=\tau_{1}=\tau_{2}=k=1$, $r_{2}=20$, given the zero initial value of the estimation parameters and initial state $x\left(0\right)={\left[1\;0\right]}^{T}$. In order to illustrate the superiority of the command-filter-based method designed for incommensurate fractional-order systems, the method proposed in our paper was adopted in Case 1, and the backstepping control approach proposed in [20] was utilized in Case 2. Figure 9 illustrates the reference signal, system output, and tracking error, Figure 10 and Figure 11 show the estimation error of the upper bound of approximation error D and unknown control coefficient b, respectively, Figure 12 depicts the actual control input, and the specific simulation results are summarized in Table 1.*


The tracking error in Case 1 had faster convergence speed and better control performance. According to the data in Table 1, the norm of the tracking error in Case 1 was 37.0% less than that in Case 2, and the norm of the control input in Case 1 was 12.5% lager than that in Case 2. Although the control cost slightly increased, the improvement in control effectiveness was even greater. Repeatedly calculating the fractional-order derivative of virtual control variables was avoided in Case 1. The issue of the explosion of computational complexity in the traditional backstepping design process was solved with the method proposed in this paper.

Parameter adjustment is a significant and difficult work in practical application. Next, we discuss the problem of parameter selection via the following simulation results. The control variate method was adopted, and one parameter was changed each time. Table 2, Table 3 and Table 4 show the influence of parameters c,G,k,andη on control performance, respectively.

According to the results in Table 2, Table 3 and Table 4, the following conclusions could be obtained.

1.As the c parameter increases, the overall tracking error decreases, the tracking effect improves, and the control cost correspondingly increases. Especially in the transitional process, this often caused the output of the system to overshoot and oscillate. If this performance is required, a differentiator or other methods could be utilized to arrange the transitional process.2.The G parameter does not have a linear relationship with the tracking error. In these simulation results, tracking performance was the best when G was 0.8. The adaptive command-filter dynamic surface control combined with intelligent optimization algorithms to tune parameters is a promising future research direction.3.As the k parameter increases, the norm of the tracking error decreased, and tracking performance improves. The estimation effect of the bound of approximation error improves, while the estimation effect of the unknown control coefficient worsens.

## 5. Conclusions

In this article, a command-filter control scheme of incommensurate nonlinear uncertain FOSs was studied. An adaptive fuzzy logic system was employed to estimate the system uncertainty. The FDM was introduced to analyze the stability of closed-loop systems via the indirect Lyapunov method. For the sake of eschewing the dimension explosion phenomenon, command-filter technology was utilized in the backstepping process. All the signals in closed-loop systems could converge to a minute neighborhood of equilibrium points.A numerical example indicated the effectiveness and advantages of the proposed control scheme. In future work, the predefined performance of tracking error will be taken into consideration in the proposed method. Fixed time command-filter-based dynamic surface control for incommensurate fractional-order systems is a future research hotspot.

## Figures and Tables

**Figure 1 entropy-25-00893-f001:**
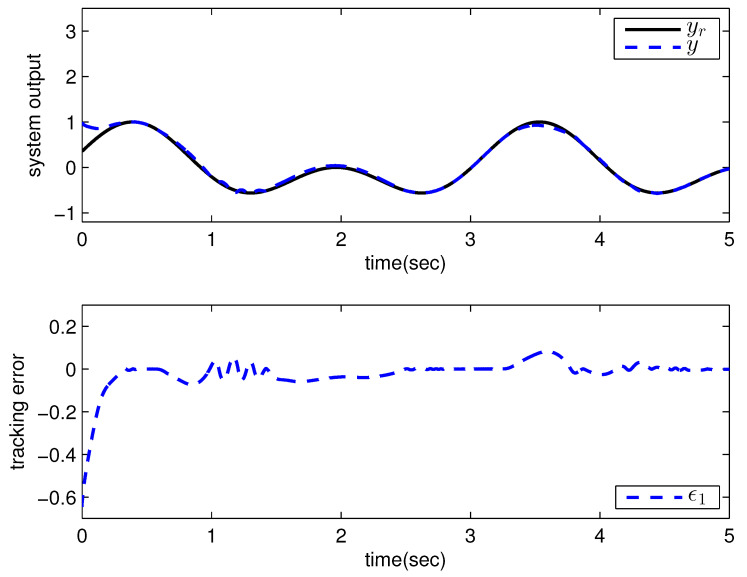
Reference signal, system output, and tracking error with the method proposed in Theorem 1.

**Figure 2 entropy-25-00893-f002:**
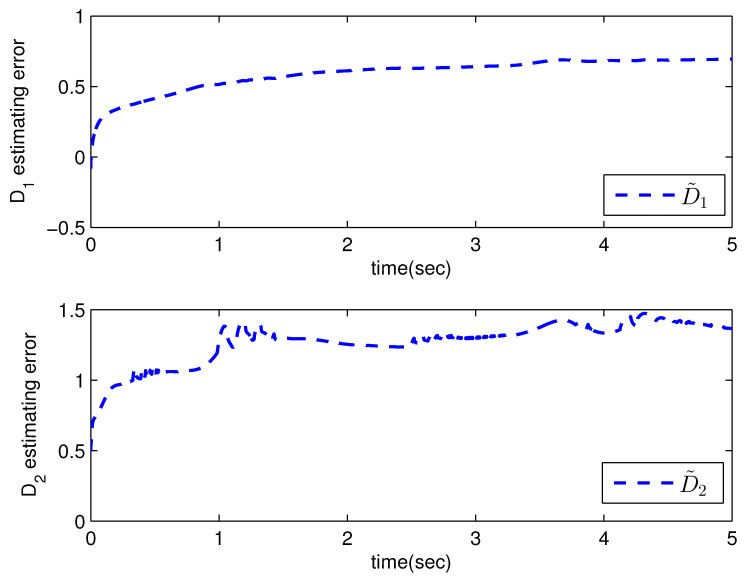
The estimation error of D with the method proposed in Theorem 1.

**Figure 3 entropy-25-00893-f003:**
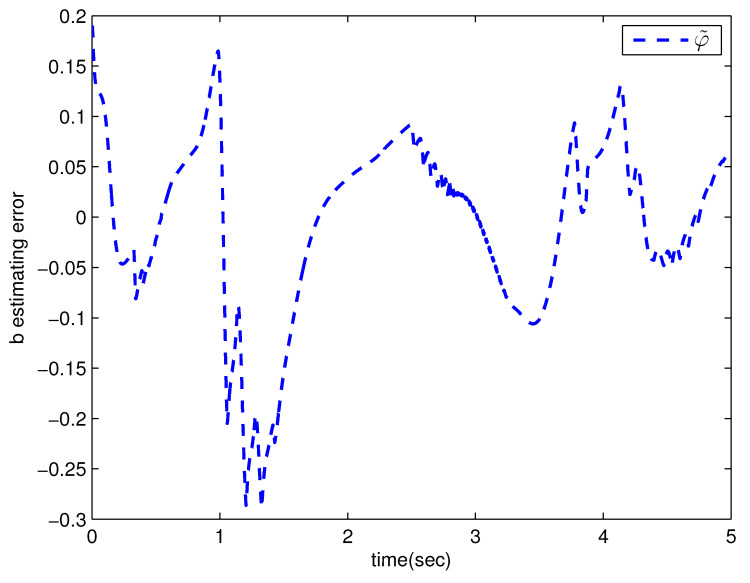
The estimation error of b with the method proposed in Theorem 1.

**Figure 4 entropy-25-00893-f004:**
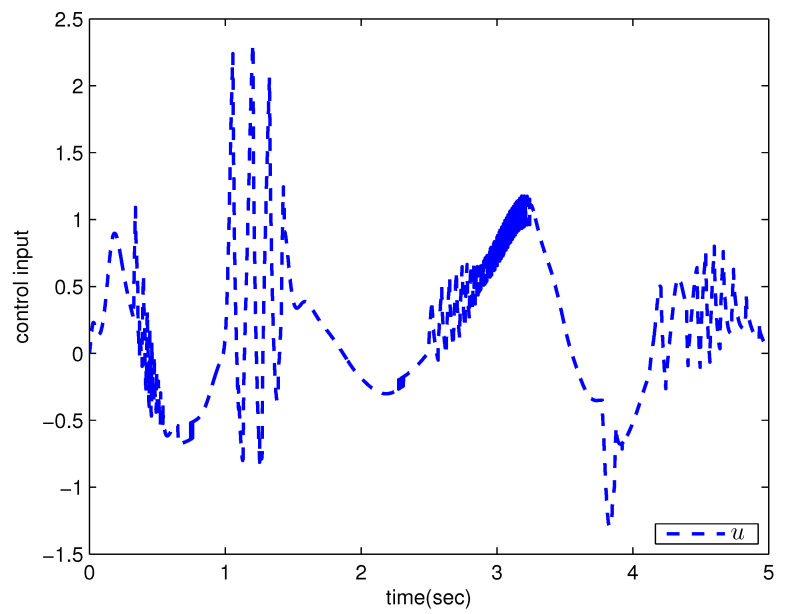
Actual control input with the method proposed in Theorem 1.

**Figure 5 entropy-25-00893-f005:**
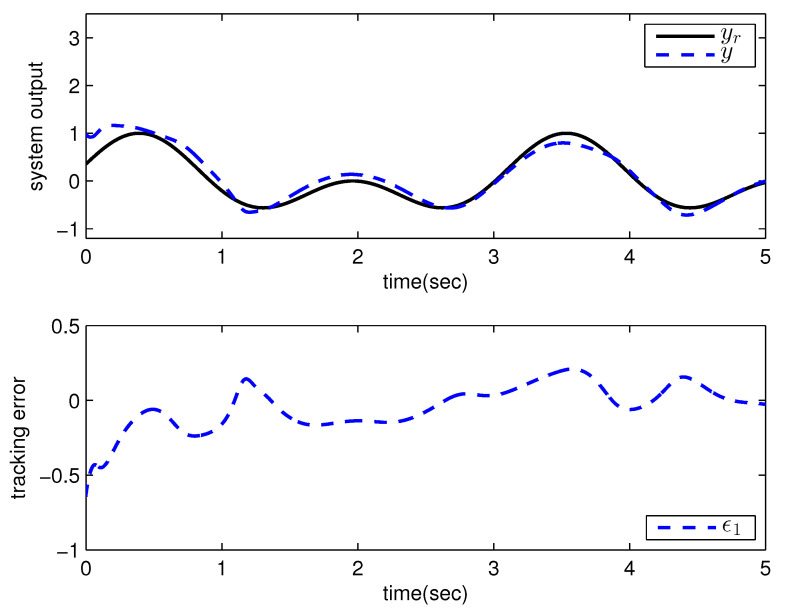
Reference signal, system output, and tracking error with the method proposed in Theorem 2.

**Figure 6 entropy-25-00893-f006:**
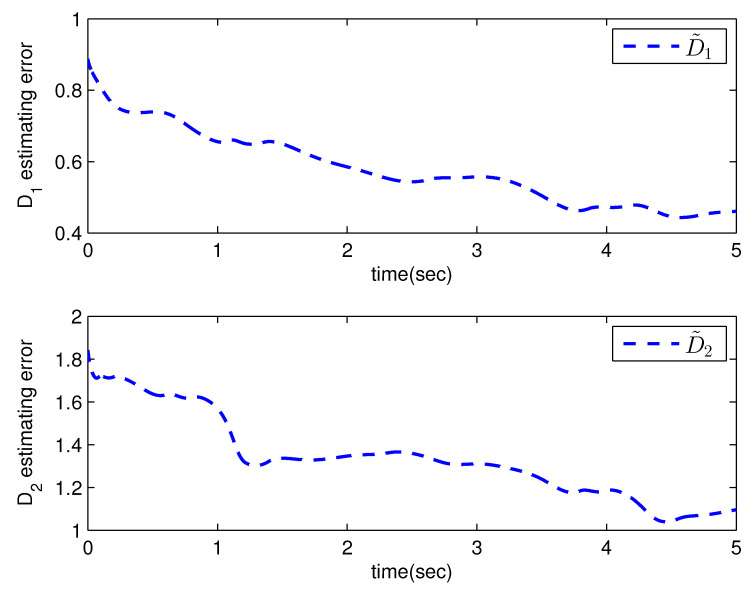
The estimation error of D with the method proposed in Theorem 2.

**Figure 7 entropy-25-00893-f007:**
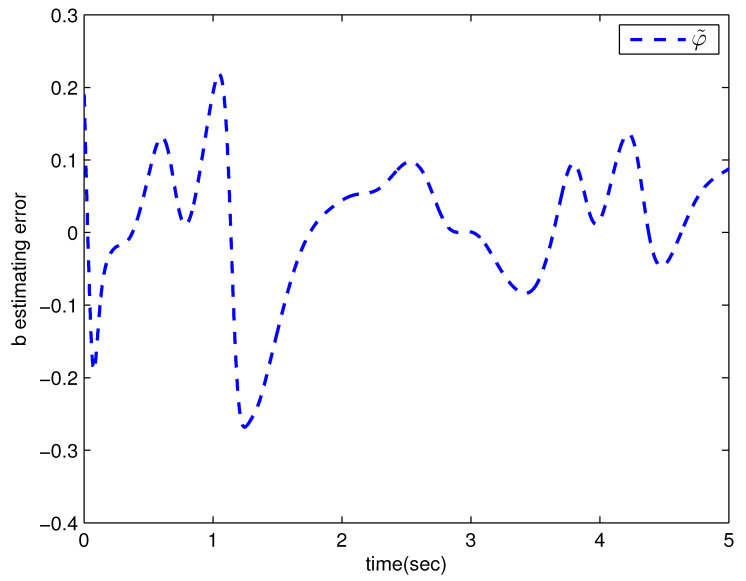
The estimation error of b with the method proposed in Theorem 2.

**Figure 8 entropy-25-00893-f008:**
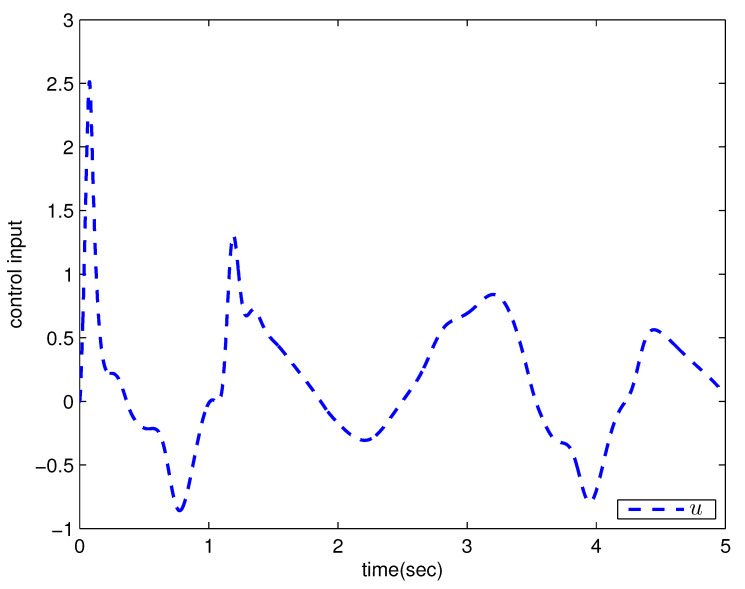
Actual control input with the method proposed in Theorem 2.

**Figure 9 entropy-25-00893-f009:**
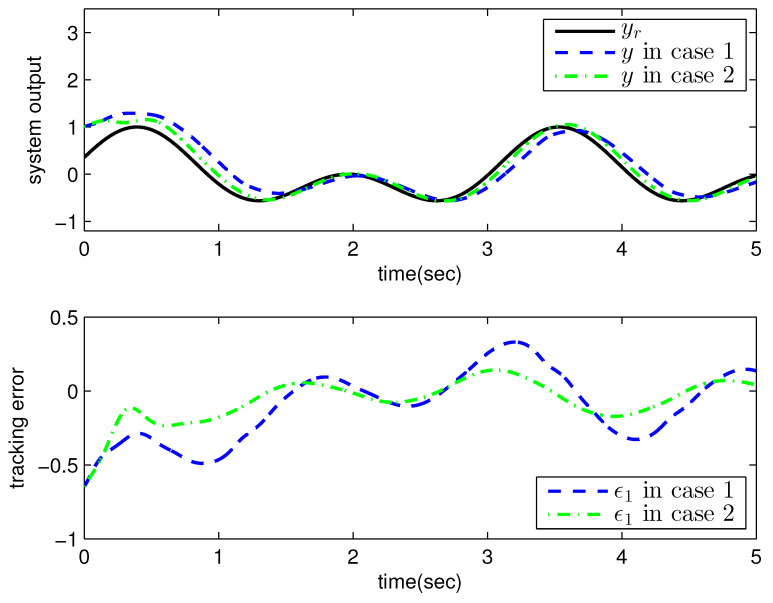
Reference signal, system output, and tracking error in Example 2.

**Figure 10 entropy-25-00893-f010:**
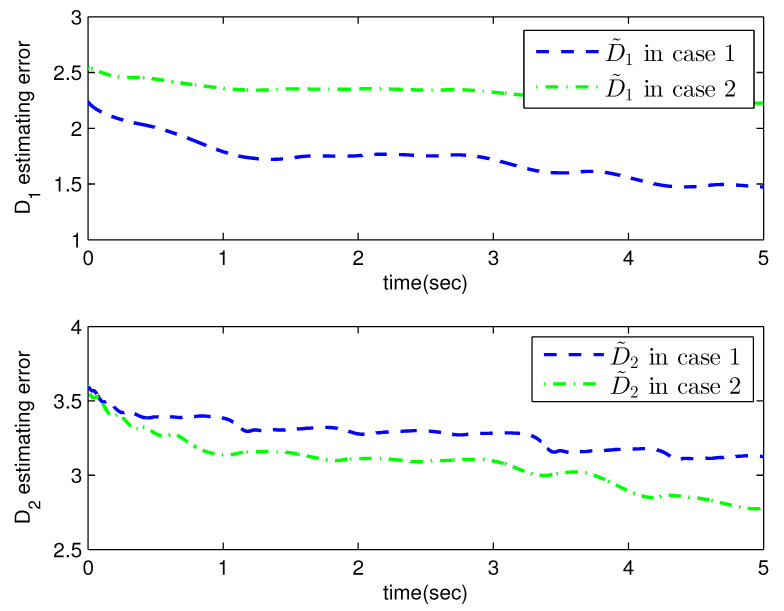
The estimation error of D in Example 2.

**Figure 11 entropy-25-00893-f011:**
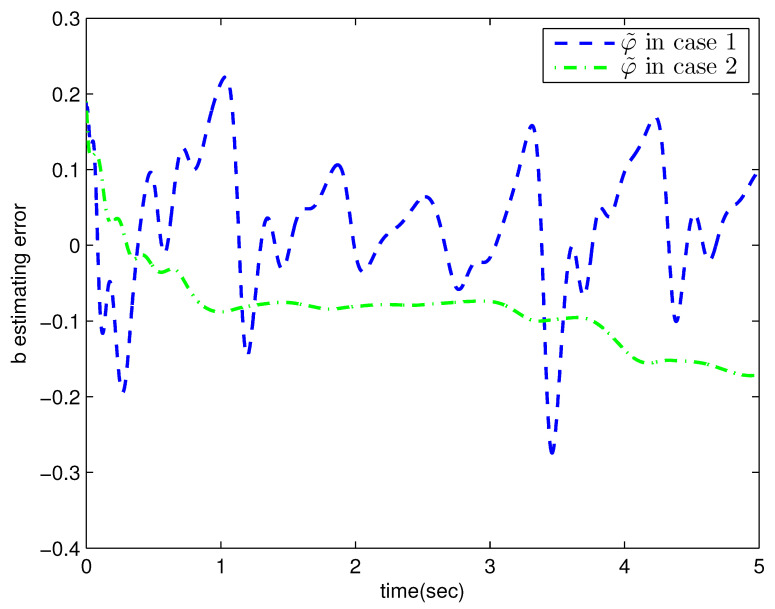
The estimation error of b in Example 2.

**Figure 12 entropy-25-00893-f012:**
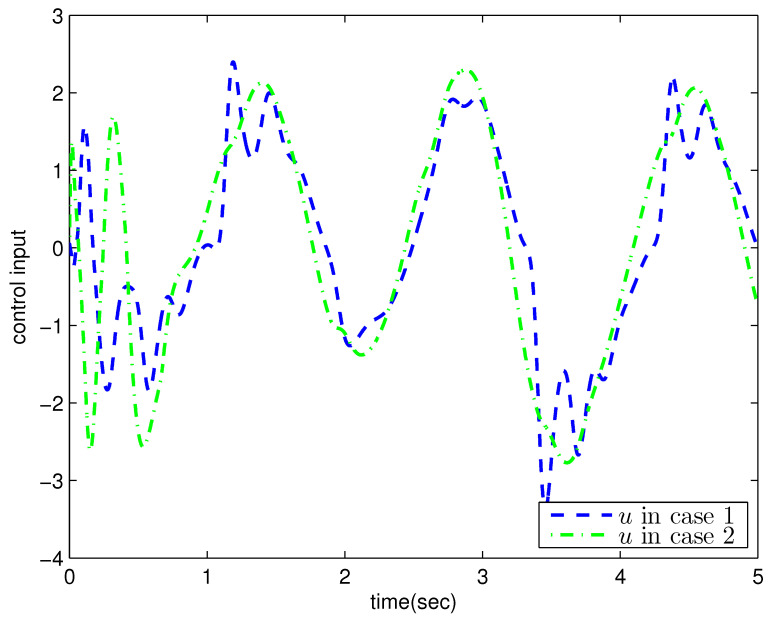
Actual control input in Example 2.

**Table 1 entropy-25-00893-t001:** Comparison of the control performance of the two methods in Example 2.

	${∥\epsilon \left(t\right)∥}_{2}$	$∥{\tilde{D}}_{1}{∥}_{2}$	$∥{\tilde{D}}_{2}{∥}_{2}$	$∥\tilde{\varphi }{∥}_{2}$	${∥u\left(t\right)∥}_{2}$
case 1	11.889	167.56	221.70	7.2698	106.82
case 2	18.867	124.94	235.26	6.6809	93.505

**Table 2 entropy-25-00893-t002:** Changing parameter c.

	${∥\epsilon \left(t\right)∥}_{2}$	$∥{\tilde{D}}_{1}{∥}_{2}$	$∥{\tilde{D}}_{2}{∥}_{2}$	$∥\tilde{\varphi }{∥}_{2}$	${∥u\left(t\right)∥}_{2}$
c = 5	20.650	150.99	238.68	139.87	97.527
c = 10	14.611	162.13	192.21	28.065	100.84
c = 15	11.889	167.56	221.70	7.2698	106.82
c = 20	10.360	168.65	202.19	8.3325	115.75

**Table 3 entropy-25-00893-t003:** Changing parameter G.

	${∥\epsilon \left(t\right)∥}_{2}$	$∥{\tilde{D}}_{1}{∥}_{2}$	$∥{\tilde{D}}_{2}{∥}_{2}$	$∥\tilde{\varphi }{∥}_{2}$	${∥u\left(t\right)∥}_{2}$
G = 0.9	12.175	167.02	219.15	7.1657	106.41
G = 0.8	11.889	167.56	221.70	7.2698	106.82
G = 0.7	12.155	168.29	224.54	7.4307	106.66
G = 0.6	12.126	168.92	227.02	7.5888	106.84

**Table 4 entropy-25-00893-t004:** Changing parameter k.

	${∥\epsilon \left(t\right)∥}_{2}$	$∥{\tilde{D}}_{1}{∥}_{2}$	$∥{\tilde{D}}_{2}{∥}_{2}$	$∥\tilde{\varphi }{∥}_{2}$	${∥u\left(t\right)∥}_{2}$
k = 5	11.501	90.087	112.93	98.327	113.01
k = 1	11.889	167.56	221.70	7.2698	106.82
k = 0.5	12.640	189.28	247.04	3.3027	107.19
k = 0.1	13.781	214.46	261.48	7.4787	109.44

## Data Availability

All comparative data indicators can be found in the references.

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
