# Peer review of "Fuzzy Adaptive Command-Filter Control of Incommensurate Fractional-Order Nonlinear Systems"

_entropy, 2023, doi:10.3390/e25060893_

Round 1

Reviewer 1 Report

The research focuses on command filter control design of nonstrict-feedback incommensurate fractional order systems. Overall is a great work.

Some comments for improvement as follow.

1. Compare the proposed control with existing similar controller for performance verification.

2. To highlight the significant of error tracking improvement. E.g. error analysis / percentage error tracking improvement.

3. Little up to date references within the latest 5 years.

Some grammatical error detected in past tense usage and starting sentences with 'And...'

Reviewer 2 Report

This paper is concerned withthe command filter control of nonstrict-feedback incommensurate fractional order systems. To deal with the dimension explosion phenomenon in the backstepping process, the authors have designed a fractional order filter and apply the command filter control technique.

This paper contains some interesting results. Please find the following comments that may help improve the paper.

1) It is suggested to highlight the challenges of the investigated problem.

2) Assumption 1 can be restrictive. Is that possible to consider a more general case? Please discuss it.

3) The backstepping technique is key for the whole paper, but is there some shortcomings for this method? Please discuss it.

4) Please double check (47).

5) In simulartion, the results seems good. But I would expect some more discussions with the obtained results.

6) Please confirm the inequality of (49).

7) The language of this paper can be further improved.

8) The conclusion section can be enhanced by discussing some future directions such as extending the asymptotic convergence to the fixed-time counterpart. The authors may refer to Collective behaviors of mobile robots beyond the nearest neighbor rules with switching topology.

9) Overall, this paper is organized well, and a revision is suggested in this round.

As above.

Reviewer 3 Report

The paper id devoted to design the control law based on backstepping method and fuzzy adaptive filter. The introduction contains a wide review of the literature and a clear problem formulation. The paper contains preliminary information, which facilitates the understanding of the material. The text is written consistently, all conclusions are correct and understandable.

There are a few comments that will improve the presentation of the paper:

1.      How to guarantee the positiveness of the function (21)? Since (21) contains coefficients that depend on sinusoids.

2.      The virtual controller (13) depends on sgn function. What happens to chattering in a closed system?

Round 2

Reviewer 2 Report

The authors have well addressed all my comments. This paper can be accepted now.

The authors have well addressed all my comments. This paper can be accepted now.